# E-Learning versus Face-to-Face Methodology for Learning Antimicrobial Resistance and Prescription Practice in a Tertiary Hospital of a Middle-Income Country

**DOI:** 10.3390/antibiotics11121829

**Published:** 2022-12-16

**Authors:** Paulina Isabel Armas Freire, Gilberto Gambero Gaspar, Jeannete Zurita, Grace Salazar, Jorge Washington Velez, Valdes Roberto Bollela

**Affiliations:** 1Medical School, Central University of Ecuador, Quito 170129, Ecuador; 2Infection Control Service, University Hospital of Ribeirão Preto Medical School, University of São Paulo, Ribeirão Preto 14049-900, Brazil; 3Biomedical Research Unit, Zurita & Zurita Laboratories and Medical School, Pontifical Catholic University of Ecuador, Quito 170104, Ecuador; 4Infection Service, Oncologic Solca Hospital, Quito 170138, Ecuador; 5Division of Education and Research, Hospital de Especialidades Eugenio Espejo, Central University of Ecuador, Quito 170136, Ecuador; 6Department of Internal Medicine, Ribeirão Preto Medical School, University of São Paulo, Ribeirão Preto 14049-900, Brazil

**Keywords:** antimicrobial prescription, antimicrobial resistance, E-learning, continuous professional development

## Abstract

Background: Antimicrobial resistance is a growing health problem worldwide. One strategy to face this problem in a reasonable way is training health personnel for the rational use of antimicrobials. There are some difficulties associated with medical staff to receiving training with E-learning education, but there is a lack of studies and insufficient evidence of the effectiveness of this method compared to face-to-face learning. Methods: An educational intervention on antimicrobial resistance (AMR) and antimicrobial prescription practice (APP) was designed and implemented using two approaches: face-to-face and E-learning among physicians of the intensive care unit (ICU) and internal medicine ward (IMW) at Eugenio Espejo Hospital in Quito. Modalities of interventions were compared to propose a strategy of continuous professional development (CPD) for all hospital staff. An interventional study was proposed using a quasi-experimental approach that included 91 physicians, of which 49 belong to the IMW and 42 to the ICU. All of them received training on AMR—half in a face-to-face mode and the other half in an asynchronous E-learning mode. They then all participated on APP training but with switched groups; those who previously participated in the face-to-face experience participated in an E-learning module and vice-versa. We evaluated self-perception about basic knowledge, attitudes and referred practices towards AMR and APP before and after the intervention. A review of medical records was conducted before and after training by checking antimicrobial prescriptions for all patients in the ICU and IMW with bacteremia, urinary tract infection (UTI), pneumonia, and skin and soft tissue infection. The study received IRB clearance, and we used SPSS for statistical analysis. Results: No statistically significant difference was observed between the E-learning and the face-to-face methodology for AMR and APP. Both methodologies improved knowledge, attitudes and referred practices. In the case of E-learning, there was a self-perception of improved attitudes (*p* < 0.05) and practices (*p* < 0.001) for both AMR and APP. In face-to-face, there was a perception of improvement only in attitudes (*p* < 0.001) for APP. In clinical practice, the use of antimicrobials significantly improved in all domains after training, including empirical and targeted treatment of bacteremia and pneumonia (*p* < 0.001) and targeted treatment of UTI (*p* < 0.05). For the empirical treatment of pneumonia, the mean number of antibiotics was reduced from 1.87 before to 1.05 after the intervention (*p* = 0.003), whereas in the targeted management of bacteremia, the number of antibiotics was reduced from 2.19 to 1.53 (*p* = 0.010). Conclusions: There was no statistically significant difference between the effect of E-learning and face-to-face strategy in terms of teaching AMR and APP. Adequate self-reported attitudes and practices in E-learning exceed those of the face-to-face approach. The empiric and targeted use of antimicrobials improved in all reviewed cases, and we observed an overall decrease in antibiotic use. Satisfaction with training was high for both methods, and participants valued the flexibility and accessibility of E-learning.

## 1. Introduction

Antimicrobial resistance (AMR) is an unresolved public health problem worldwide that requires the intervention of various sectors, including education [1]. Antibiotics, as a public good and non-renewable collective, require rational use to preserve their efficacy because inappropriate use adds considerable costs to patient care and increases morbidity and mortality [2]. Over 50% of all medicines are prescribed inappropriately [3,4], and it is estimated that antibiotics represent 30% of a hospital’s pharmacy cost [5]. Most inappropriate prescriptions result from inadequate antibiotics at incorrect doses, non-optimal concentrations in the focus of infection, durations that are too long or too short, and a lack of de-escalation to the oral route [4,6].

In a study by Sandoval et al., (2018), a frequency of medical prescription error of 51.2% was determined in an Ecuadorian emergency service, with antibiotics being the therapeutic group with the most prescription errors (35.8%; *p* = 0.003). In Ecuador, there are no treatment guides or training courses on antimicrobial resistance (AMR) and antimicrobial prescription practice (APP) provided by the state, which aggravates the problem.

Physicians′ knowledge of antibiotic use has been identified as a key factor affecting individual prescribing behavior in clinical practice [5,6,7,8]. For this reason, it is recommended that improvement programs focus on these issues [7], and in several investigations, the same prescriber demands regular training [8,9,10,11]. E-learning tools (virtual distance education through electronic channels using digital applications) can be integrated into the existing training tools for the continuous strengthening of the prescribing team [12,13,14,15], also allowing the massive dissemination of knowledge at a lower cost in comparison with other methodologies [16,17].

Previous surveys reported that online methods are likely effective as alternative methods for training health professionals in clinical practice, but further evidence is required [18]. E-learning has been successfully applied to AMR and APP training [14,19,20], but no study has contrasted the effect on these subjects and the methodology in learning in a middle-income country such as Ecuador.

The aim of this study was to design and evaluate the short-term impact of an educational intervention on AMR and APP based on E-learning and face-to-face approaches with doctors of the intensive care unit (ICU) and internal medicine ward (IMW) at Eugenio Espejo Hospital (HEE) in Quito, Ecuador.

## 2. Results

### 2.1. Assessment of Learning

The AMR and APP groups had 45 and 46 doctors, respectively. Among the 91 participants, 54.9% were women; 53.8% worked in the IMW and 46.2% in the ICU. In terms of position, 31.9% were specialist doctors, 45% were postgraduate doctors and 23.1% were residents. The mean age was 32.4 ± 8.6 years, with 3.9 ± 5.1 years of clinical practice. Most (80.2%) participants stated that they use the Internet or another online platform as a regular source of information. Almost all (98.9%) recognized the importance of antimicrobial APP training and the use of clinical practice guides. Slightly fewer (96.7%) also considered the contribution of infectious diseases physicians and microbiologists useful, whereas 83.5% reported that they also look for information on the Internet. On the other hand, previous clinical experience (48.4%) and the contribution of peers from the same specialty (23.1%) were not considered very useful.

The participants filled out the questionnaire before and after the training. A total of 45 and 46 participants completed each module in each modality, respectively.

When comparing the learning methodologies, it was observed that there were no significant differences between the post-training results (in knowledge, attitudes and referred practices) obtained using the two investigated methodologies (face-to-face and E-learning). In both modules, there was not a difference in the expected outcomes between the E-learning and face-to-face learning approaches.

On the other hand, when comparing before and after training within each module for both modalities, there seems to be a post-training improvement in knowledge, attitudes and practices in all aspects evaluated. In the AMR module, attitudes and practices improved significantly with the E-learning methodology (*p* < 0.05). (Table 1). In the APP module, attitudes (*p* < 0.001) and practices (*p* = 0.037) improved significantly with the E-learning methodology, and attitudes improved significantly compared with the face-to-face methodology (*p* = 0.001). (Table 1). The improvement was not significant in knowledge.

### 2.2. Assessment of Antibiotic Appropriateness

In the second part of this study, the prescription practice was evaluated in 257 patients of the intervened services before (122) and after (135) the intervention, of which 25 died during the empirical or targeted treatment and were therefore registered as non-assessable.

We included patients with bacteremia (131), pneumonia (72), urinary tract infection (37), and skin and soft tissue infection (17). All patients were hospitalized (Table 2 and Table 3).

In patients with soft tissue and skin infections, it was observed that most of were treated with surgical cleaning and did not require antibiotic treatment, especially after the intervention. In the empirical treatment of pneumonia, it was observed that in post intervention there were more cases in which it was justified not to administer the treatment (15.6% versus 40%; *p* = 0.024; OR: 0.27, 95% CI 0.08–0.87). Likewise, the justification for not giving antibiotic treatment in the empirical and targeted treatment of bacteremia was increased; however, it was not significant.

The coverage of the microorganism improved significantly in the empirical treatment of bacteremia (66.7% before versus 83.1% after; *p* < 0.001), and it was also evident that before the intervention, there was a greater risk of having poor antibiotic coverage (OR = 10.43 (2.28–47.61)). Targeted pneumonia treatment also improved (59.4% before versus 82.5% after; *p* = 0.018), however, it was worse in empirical treatment (71.9% before versus 57.5% after; *p* = 0.034), possibly related to suspected COVID-19 (Table 2 and Table 3).

In the use of antibiotics with a broader spectrum than necessary, there was a decrease in all types of infection evaluated, except for the empirical treatment of bacteremia, possibly related to the critical condition of these patients, with a significant reduction in empiric treatment (50% before versus 15% after; *p* = 0.004) and targeted (50% before versus 25% after; *p* = 0.019) of pneumonia, and in targeted treatment of urinary tract infection (43.8% before versus 9.5% after, *p* = 0.045) (Table 2 and Table 3).

The adequacy of the prescription in terms of the dose and duration of the antibiotic treatment increased in almost all the pathologies evaluated, with significant improvement in the empiric treatment of bacteremia (51.5% before versus 76.9% after; *p* < 0.001) and a higher risk of being inadequate before the intervention (OR= 6.61 (2.46–17.74)). Here, it was observed that the dose and duration did not improve in the empiric treatment of pneumonia (65.6% before versus 60% after; *p* = 0.004) (Table 2 and Table 3).

The rational use of antimicrobials significantly improved the empirical and targeted treatment of all types of infections, with a significant improvement in the treatment of bacteremia in the empirical (54.5% before versus 92.3% after; *p* < 0.001; OR= 10.0 (3.55–28.09) and in the targeted treatment (51.5% before versus 80% after; *p* = 0.001; OR= 5.86 (2.16–15.89). Before the intervention, there was an increased risk of inappropriate use of antibiotics in both cases. In pneumonia, empiric (62.5% before versus 100% after; *p* < 0.001) and targeted treatment (34.4% before versus 90% after; *p* < 0.001) improved. Additionally, it was found that the risk of inappropriate rational use was 26 times higher in the targeted treatment before the intervention. Finally, in the targeted treatment of urinary tract infection, rational use was also significant (81.3% before versus 100% after, *p* = 0.038) (Table 2 and Table 3).

Furthermore, in the 257 analyzed cases, it was observed that there was a decrease in the total number of antimicrobials prescribed after the intervention. In the case of empirical management, a statistically significant difference was observed in the case of patients with pneumonia; the mean number of antibiotics used was 1.87 before the intervention, decreasing to 1.05 after (*p* = 0.003), whereas under targeted management, this trend was evidenced in patients with bacteremia, for whom the mean before the intervention was 2.19 antibiotics, falling to 1.53 with a *p* of 0.010 (Table 4).

### 2.3. Satisfaction Evaluation

The level of satisfaction of the participants with the course was excellent; 79.12% said that it exceeded their expectations, and 20.88% placed it within their expectations. When asked what they liked the most, 22% said that the topics were useful and pertinent, and 24.2% indicated that the presentations were educational.

When asked about what could be improved, 16.5% said that the course could include more clinical cases; 13.2% said that the course could be longer, given the importance of the topic; and 9.9% suggested that the course could be repeated more frequently.

## 3. Methods

### 3.1. Study Design and Population

This study follows a quasi-experimental approach based on an educational intervention that was conducted in a public tertiary hospital in Quito, Ecuador. The hospital has a department of microbiology but does not have infectious disease doctors. The hospital has 434 beds and 592 doctors (263 specialist physicians, 197 residents and 132 postgraduates).

A total of 141 internal medicine and intensive care physicians were invited to the training and were randomly assigned into two groups of 70 and 71 participants, respectively. Within the framework of a crossover study, one group received AMR training by face-to-face methodology and training in APP by E-learning methodology; the other group received AMR training by E-learning methodology and APP by face-to-face methodology. In the end, there were 45 and 46 participants in the first and second group, respectively, who signed an informed consent, approved the training, and completed the questionnaire. In group 1, there were 15 specialists, 12 residents and 20 postgraduates. In group 2, there were 14 specialists, 9 residents and 21 postgraduates. All of them were medical graduates with a professional license to practice. Regarding the units to which they belonged, 49 worked in internal medicine, and 42 worked in intensive care, equally distributed in groups 1 and 2.

In the first phase of the study, 91 specialist physicians, residents and postgraduates from the ICU and IMW services were trained, and knowledge, attitudes and referred practice of AMR and APP were evaluated by means of a survey before and after the training.

In the second phase of the study, the prescription practice was evaluated in the services that were trained through the review of all cases of bacteremia, pneumonia, urinary tract infection (UTI), and skin and soft tissue infections, with 122 cases before and 135 cases after of the training. To this end, all microbiology reports and digital medical records of patients attending the IMW and ICU of the HEE over 8 weeks throughout 2021 (1–31 May; 1–31 July) were included.

### 3.2. Data Description

In the first phase of this study, the AMR course was based on the state of the art of the subject at a local and global level. The course on APP was based on the WHO course entitled Antimicrobial Stewardship: A Competency-Based Approach [21]. To guide the empirical therapeutic decision, the updated susceptibility chart for the second half of 2020 was declared by the microbiology team, and the PAHO Infectious Diseases 2021–2022 e-book was delivered [22].

The trainers were professionals with extensive experience in research and teaching in their field, with experts in each topic to be developed; five classes were created for both AMR and APP, all approximately 45 min long. All the contents were placed on the Moodle platform in an audiovisual format for the E-learning modality, together with a supporting bibliography.

The trainers used the same content in the face-to-face training for both groups. The duration of each face-to-face meeting was approximately four hours. Those in the face-to-face group did not have access to the E-learning material. The study design is shown in Figure 1.

The educational intervention highlighted that administration of antimicrobials should follow adequate clinical assessment for the initial diagnosis, therapeutic decision and the need for constant re-evaluation and review of laboratory data to adequately identify antimicrobials in a subsequent evaluation.

In the second phase of the study, empirical choice of antimicrobial was based on the severity of the infection, characterization of disease compatibility with an infection, bacterial resistance patterns in the community or hospital and specific factors related to the patient and their disease.

For the informed choice of the targeted treatment after clinical re-evaluation based on microbiological report, the antimicrobial spectrum, adverse effects, dose, route and duration of the therapy were taken into consideration. The decision can be made to maintain the therapy, to escalate, to de-escalate or to interrupt antimicrobial treatment depending on each case.

### 3.3. Assessment of Methodology of Learning

In the first phase of the study, the central questions of the questionnaires were adapted from questions applied by García Coralith et al., (2011) and Nair Mohit et al., (2019), who evaluated perceptions, knowledge, attitudes and practices with respect to AMR and APP among health personnel [10,23]. The questionnaire completed to all participants was the same, and it was validated by 7 experts (2 infectious disease physicians and specialists in tropical diseases, 2 clinical pathologists, 1 microbiologist and 2 epidemiologists) who thoroughly reviewed the questionnaire and concluded that it measures the traits of interest by evaluating each question: adequacy, relevance, language comprehension, adaptability to the medium and accuracy of the Spanish translation. A pilot test was carried out on 30 subjects, and a Cronbach′s alpha of 0.79 was obtained. The questionnaire is available as Appendix A.

The questions collect information on the previous knowledge to introduce the participant to the attitude questions aimed at evaluating the willingness to be trained on the subject; on the other hand, the referred practice questions evaluate through clinical cases that require prior knowledge for decision making. The training took place in June 2021, and the questionnaire was completed before and after the training.

### 3.4. Evaluation of Antibiotic Appropriateness

In the second phase of the study, to identify patients who had an infection and who were hospitalized in the IMW or ICU, we started with the list of all positive reports of blood cultures, tracheal secretions, urine cultures and cultures of skin and soft tissue secretions that were processed in the microbiology laboratory of the hospital between May and July 2021. The cases to be studied were identified from the positive microbiological reports. In each case, the empirical and targeted antimicrobial prescription was investigated in the medical records, along with other data that would enable evaluation of the adequacy of the prescription of antimicrobial therapy.

Two experts in infectious disease and clinical microbiology assessed antibiotic appropriateness. The experts used epidemiological information from the local bacterial resistance card to assess empirical treatment and the PAHO Infectious Diseases 2021–2022 guide for targeted treatment of infectious diseases; the hospital does not have guidelines from the institution itself. Empirical or targeted treatment was not prescribed or continued in case cases in which the patient′s death was recorded as “Not evaluable”. The appropriate procedures and cases in which it was justified that the patient did not require antimicrobial treatment were recorded as “adequate”.

We analyzed all registered cases, and for each case, both in the empirical and targeted treatment, the following variables were adapted and included:Indication to start treatment: decision to give antibiotic treatment when the patient had an infection that justified it;Indication not to treat: clinical and laboratory justification of the decision not to treat a patient who did not present with an infection;Coverage of the microorganism: the antibiotic treatment covered the microorganism suspected by epidemiology in the empirical treatment or one identified by culture in the case of targeted treatment;Antibiotic spectrum: the used spectrum was sufficient for the suspected or reported microorganism;Greater spectrum than necessary: the antimicrobial spectrum was greater than necessary for the suspected or reported microorganism;Dose and duration of treatment: whether the dose and duration of the treatment administered were adequate or not;Rational use of antibiotics corresponds to the sum of the indication to give treatment and not to give it in case of clinical and laboratory justification, adequate coverage of the microorganism, the antibiotic spectrum used and the dose and duration of treatment.

### 3.5. Data Analysis and Sample Size

In the first phase of the study, to determine ratios (including prescription appropriateness) with a 95% certainty and an error of 0.5%, we calculated that we required a total of 69 participants to be trained. A total of 91 doctors were trained.

All participants were physicians who were a permanent or temporary part of the ICU or IMW service at the HEE during the educational intervention from May to July 2021. All participants signed an informed consent, completed all face-to-face and virtual training and delivered complete evaluations.

In the second phase included all patients with infection and a positive microbiological report to assess the observed prescribing practice.

### 3.6. IRB Approval

The Human Research Ethics Committee Beings of the Pontificia Universidad Católica del Ecuador approved the study (EO-01-2020).

## 4. Discussion

Low- and middle-income countries are at the epicenter of AMR as a growing public health threat [24,25]. Government and local organizations are in dissimilar stages of implementation to address AMR and antimicrobial prescription practices [26]. In Ecuador, the implementation of antimicrobial stewardship has not been achieved in most secondary and tertiary hospitals, and AMR continues to increase [27]. In Ecuador, there is no specialization in infectious diseases, which are the responsibility of internists. During the COVID-19 pandemic, the country suffered great deficiencies in health care, in hospital infrastructure, equipment and medical personnel. In Ecuador, there is no professional qualification exam on antibiotic prescription, and continuous training on this subject for practicing professionals is non-existent.

Since the end of 2019, due to the COVID-19 pandemic, there has been an abrupt shift towards online learning at almost all levels of education around the world. The pandemic directly affected health personnel, and they never ceased to have the professional responsibility to maintain competence in clinical practice; to achieve this, one of the main strategies was online education [28]. However, not all health personnel have the time or resources to access a face-to-face training or refresher courses, and concerns remain as to whether courses offered virtually have the desired effect.

The Kerala model in India demonstrates that multiple interventions centered on education have a considerable and sustainable impact on AMR and APP [25]; as described by Singh et al., (2021), the generation of useful content and its dissemination through digital platforms can contribute generously to good prescription practice strategies [25]; the availability of educational resources is one of the benefits offered by the E-learning methodology, and this reaches a greater value when resources are limited, as is the case in the public sector of low- and middle-income countries [25].

AMR and APP of antimicrobials are topics of foremost importance in global public health that were present before the COVID 19 pandemic and will persist after it, and due to high consumption during the pandemic, it is possible that the problem will worsen [29].

These results suggest that there is no statistically significant difference between the E-learning methodology and traditional methods, which is consistent with meta-analyses by Liu et al., (2016) and Richmond et al., (2017) [18,30]. There is a study in which the impact of an online educational intervention on antibiotic resistance in a tertiary hospital was evaluated with a cross-sectional design, producing significant changes in knowledge [20]; however, on the subject of AMR and APP, no further studies have been published including a comparison of the learning methodology within the framework of a cross-sectional design, evaluating knowledge, attitudes, and related and observed practices.

In the present study, we also observed that in changing attitudes and practices, E-learning could even outperform face-to-face methodology, for which more evidence is required. However, these results correlate with the findings of Sinclair et al., (2016) [16], George et al., (2019) [31] and Fontaine et al., (2019) [32], who carried out meta-analyses on E-learning versus face-to-face learning in various topics of health interest and found that E-learning was at least as effective as the traditional approach. Interestingly, Fontaine′s analysis suggests that electronic learning environments seem particularly effective in improving the skills of professionals and students in the health field, as was the case in the present study.

It is not common to evaluate the level of satisfaction of healthcare professionals with a course, but there are some cases. For example, a study by Sahin et al., (2008) evaluated the satisfaction of dentists with an educational intervention on prescription practices [33], and a study by Chavez et al., (2020) reported on physicians [34]; however, the results achieved in the present study exceed those described in these studies.

The rational use of antimicrobials is a variable that summarizes the appropriate indication to give treatment or not to give it, the coverage of the microorganism, the spectrum used, and the dose and duration of treatment. This variable significantly improves in the empirical and targeted treatment of all types of infections, with significant improvement in the treatment of bacteremia and pneumonia, both empirical and targeted; before the intervention, there is an increased risk of inappropriate use of antibiotics. A study by Swamy et al., (2019) showed a significant improvement in the adequacy of antibiotic prescription in general (66 vs. 86%, P < 0.001) [35], representing the closest data compared to those obtained in the present study.

At the time we conducted our research at the Eugenio Espejo Hospital, there were no pharmaceutical biochemists who evaluating the defined daily dose (DDD) or monitoring antibiotic consumption by any other method. As DDD was not available in this study, only the average use of antibiotics per patient in the pathologies evaluated is presented. We observed a decrease in the total number of antimicrobials after the intervention. For the empirical treatment of pneumonia, the mean number of antibiotics decreased from 1.87 before to 1.05 after the intervention (*p* = 0.003), whereas in the targeted management of bacteremia, it decreased from 2.19 to 1.53 (*p* = 0.010). We can compare these data with those from studies by Swamy et al., (2019), who showed a reduction in the mean number of antibiotics used per person (4.41 vs. 3.86, P < 0.05) after an educational intervention in a tertiary hospital [35]; however, no study has discriminated between empirical and targeted treatment or between infections from different foci as performed in the present study.

It is interesting to note that there are also non-quantifiable changes that could be observed in a review of medical records after the training; we observed, for example, that physicians began to include in the medical record the justification for the prescription of the antimicrobial or, failing that, the justification for the non-prescription, shortened treatment, monotherapy in the targeted treatment and the clinical effects of the course of treatment. In addition, the de-escalation of vancomycin to linezolid in SAMS, etc., was recorded. These data, although important in medical competencies, are not quantifiable and have not been recorded in other studies.

Given the differences between E-learning and face-to-face strategies in the topics addressed and considering that one-fifth of the participants showed interest in reviewing a greater number of clinical cases and greater interactivity on the Moodle platform, a proposal for blended learning with a flipped classroom was generated for the implementation of a continuous training program that includes all the prescribing staff of the hospital.

In summary, no similar publications on AMR and APP have been published that evaluate the knowledge, attitudes and practices referred to with two educational methodologies, in addition to evaluating participant satisfaction and scaling the evaluation to the effects produced by the training in the observed practice, as was the case in this research. We evaluated the three levels of the Kirkpatrick model [36], although other publications on infection control with significant results have used the same model [37].

This study has limitations, as it is based on a small number of professionals in a single hospital in Ecuador. We did not perform a long-term evaluation and could not provide feedback to all the prescribing professionals because some of them rotated to other services; we believe that this should be included in subsequent studies. We also included a small number of patients with skin and soft tissue infections, which did not allow us to obtain significant results related to this topic.

The implementation of an educational intervention can have a positive effect in raising awareness among healthcare personnel that AMR is a problem that we can influence in our daily work and that good prescribing practices are among the viable solutions that we can provide in different environments, such as that of a public hospital in a middle-income country that must adapt to changes in E-learning education and take advantage of the benefits offered by this methodology.

## 5. Conclusions

There was no statistically significant difference between the effect of the E-learning and the face-to-face methodology for the learning of AMR and APP. Both methodologies improve knowledge, attitudes and referred practices. E-learning could be considered an effective option for the development of healthcare professionals with respect to AMR and APP. These interventions can have an effect on reducing the amount of antibiotics prescribed and increasing appropriate treatment.

## Figures and Tables

**Figure 1 antibiotics-11-01829-f001:**
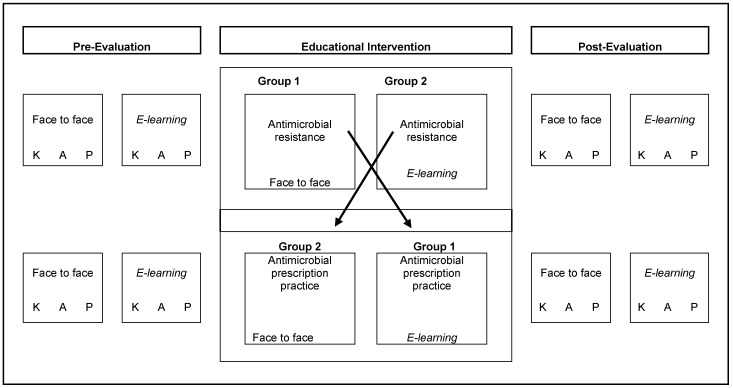
Assessment of knowledge, attitudes and referred practices before and after the educational intervention. Legend: K: knowledge; A: attitudes; P: practices.

**Table 1 antibiotics-11-01829-t001:** Effect of the training modality on knowledge, attitudes and referred practices before and after the educational intervention.

	Module: Antimicrobial Resistance								Module: Antimicrobial Prescription Practice							
	Questions	Face to Face			E-Learning				Questions	Face to Face			E-Learning			
	(Min–Max)	Median	(P25–P75)		Median	(P25–P75)		*p* *	(Min–Max)	Median	(P25–P75)		Median	(P25–P75)		*p* *
**Knowledge**	6								6							
Before training	(1–6)	5	(5–6)		5	(5–6)		0.602	(1–6)	4	(4–5)		4	(4–5)		0.63
After training	(2–6)	6	(5–6)		6	(5–6)		0.884	(2–6)	5	(4–5)		4	(4–5)		1.00
** *p *** **		0.097			0.259					0.205			0.603			
**Attitudes**	4								9							
Before training	(1–4)	4	(3–4)		3	(3–4)		0.017	(2–8)	6	(5–7)		6	(5–6)		0.89
After training	(1–4)	4	(4–4)		4	(3–4)		0.345	(3–9)	7	(5–8)		7	(6–7)		0.87
** *p *** **		0.94			0.005					0.001			<0.001			
**Referred practices**	5								7							
Before training	(0–5)	3	(2–4)		3	(2–4)		0.66	(0–7)	5	(4–5)		5	(4–5)		0.68
After training	(1–5)	3	(3–4)		4	(3–4)		0.21	(2–7)	5	(4–5)		5	(5–5)		0.572
** *p *** **		0.59			<0.001					0.521			0.037			

* Mann–Whitney U test; ** Wilcoxon W test.

**Table 2 antibiotics-11-01829-t002:** Comparison of prescription practice observed in empirical treatments in 122 patients before and 135 patients after training.

	Bacteremia						Pneumonia						Urinary Infection						Skin and Soft Tissue Infection					
	Before		After				Before		After				Before		After				Before		After			
	*n* = 66		*n* = 65				*n* = 32		*n* = 40				*n* = 16		*n* = 21				*n* = 8		*n* = 9			
	*n*	%	*n*	%	**Chi2 (*p*)**	**OR** **(95% CI)**	*n*	%	*n*	%	**Chi2 (*p*)**	**OR** **(95% CI)**	*n*	%	*n*	%	**Chi2 (*p*)**	**OR** **(95% CI)**	*n*	%	*n*	%	**Chi2 (*p*)**	**OR** **(95% CI)**
**Indication to treat**																								
No	5	7.6	9	13.8	1.349 (0.245)	1.96 (0.61–6.20)	5	15.6	16	40	**5.113 (0.024)**	**0.27 (0.08–0.87)**	6	37.5	5	23.8	0.815 (0.367)	1.92 (0.46–7.98)	1	12.5	1	11.1	0.008 (0.929)	1.14 (0.05–21.87)
Yes	61	92.4	56	86.2			27	84.4	24	60			10	62.5	16	76.2			7	87.5	8	88.9		
**Coverage of the microorganism**																								
No	17	25.8	2	3.1	**13.999 (0.001)**	**10.43 (2.28–47.61)**	4	12.5	1	2.5	**6.756 (0.034)**	4.00 (0.41–38.57)	1	6.3	1	4.8	0.932 (0.627)	1.66 (0.09–30.06)	0	0	0	0	1.195 (0.274)	
Yes	44	66.7	54	83.1			23	71.9	23	57.5			9	56.3	15	71.4			7	87.5	9	100		
NV	5	7.6	9	13.8			5	15,56	16	40			6	37,5	5	23,8			1	12.5	0	0		
**Broader spectrum than necessary**																								
No	19	28.8	9	13.8	4.988 (0.083)	2.36 (0.96–5.78)	11	34,4	18	45	**11.247 (0.004)**	**0.22 (0.06–0.76)**	4	25	11	52,4	2.824 (0.244)	0.30 (0.05–1.57)	3	37.5	8	88.9	5.031 (0.081)	0.09 (0.007–1.21)
Yes	42	63.6	47	72.3			16	50	6	15			6	37.5	5	23.8			4	50	1	11,1		
NV	5	7.6	9	13.8			5	15.6	16	40			6	37.5	5	23.8			1	12.5	0	0		
**Dosage and duration**																								
Inadequate	27	40.9	6	9.2	**17.548 (<0.001)**	**6.61 (2.46–17.74)**	6	18.8	0	0	**11.211 (0.004)**	-----	2	12.5	0	0	4.158 (0.125)	---	1	12,5	0	0	2.550 (0.279)	----
Appropriate	34	51.5	50	76.9			21	65.6	24	60			8	50	16	76.2			6	75	9	100		
NV	5	7.6	9	13.8			5	15.6	16	40			6	37.5	5	23.8			1	12,5	0	0		
**Rational use of antibiotics**																								
Inadequate	30	45.5	5	7.7	**23.851 (<0.001)**	**10.0 (3.55–28.09)**	12	37.5	0	0	**18.000 (<0.001)**	-----	3	18.8	2	9.5	0.661 (0.416)	2.19 (0.32–15.0)	0	0	0	0	—	----
Appropriate	36	54.5	60	92.3			20	62.5	40	100			13	81.1	19	90.5			8	100	9	100		
NV	0	0	0	0			0	0	0	0			0	0	0	0			0	0	0	0		
NV = not valuable.																								
**Source: database**																								
**Elaborated by the authors**																								

**Table 3 antibiotics-11-01829-t003:** Comparison of the observed practice of prescription in targeted treatments in 122 patients before and 135 patients after the training.

	Bacteremia						Pneumonia						Urinary Infection						Skin and Soft Tissue Infection					
	Before		After				Before		After				Before		After				Before		After			
	*n* = 66		*n* = 65				*n* = 32		*n* = 40				*n* = 16		*n* = 21				*n* = 8		*n* = 9			
	*n*	%	*n*	%	**Chi2 (*p*)**	**OR** **(95% CI)**	*n*	%	*n*	%	**Chi2 (*p*)**	**OR** **(95% CI)**	*n*	%	*n*	%	**Chi2 (*p*)**	**OR** **(95% CI)**	*n*	%	*n*	%	**Chi2 (*p*)**	**OR** **(95% CI)**
**Indication to treat**																								
No	12	18.2	21	32.3	3.468 (0.063)	0.46 (0.20–1.05)	8	25	7	17.5	0.0606 (0.436)	1.57 (0.50–4.92)	6	37.5	10	47.6	0.379 (0.538)	0.66 (0.17–2.48)	7	87.5	4	44.4	3.438 (0.064)	8.75 (0.73–103.82)
Yes	54	81.8	44	67.7			24	75	33	82.5			10	62.5	11	52.4			1	12.5	5	55.6		
**Coverage of the microorganism**																								
No	7	10.6	2	3.1	3.894 (0.143)	3.22 (0.62–16.57)	5	15.6	0	0	**8.046 (0.018)**	----	0	0	0	0	0.379 (0.538)	----	0	0	0	0	3.438 (0.064)	-----
Yes	38	57.6	35	53.8			19	59.4	33	82.5			10	62.5	11	52.4			1	12.5	5	55.6		
NV	21	31.8	28	43.1			8	25	7	17.5			6	37.5	10	47.6			7	87.5	4	44.4		
**Broader spectrum than Necessary**																								
No	8	12.1	6	9.2	1.808 (0.405)	1.11 (0.34–3.56)	8	25	23	57.5	**7.918 (0.019)**	**0.21 (0.07–0.67)**	3	18.8	9	42.9	**6.216 (0.045)**	**0.09 (0.01–0.73)**	0	0	4	44.4	4.776 (0.092)	-----
Yes	37	56.1	31	47.7			16	50	10	25			7	43.8	2	9.5			1	12.5	1	11.1		
NV	21	31.8	28	43.1			8	25	7	17.5			6	37.5	10	47.6			7	87.5	4	44.4		
**Dosage and duration**																								
Inadequate	18	27.3	7	10.8	**5.991 (0.050)**	**2.85 (1.03–7.89)**	3	9.4	0	0	4.905 (0.086)	---	0	0	0	0	0.379 (0.538)	---	0	0	0	0	3.438 (0.064)	-----
Appropriate	27	40.9	30	46.2			21	65.6	33	82.5			10	62.5	11	52.4			1	12.5	5	55.6		
NV	21	31.8	28	43.1			8	25	7	17.5			6	37.5	10	47.6			7	87.5	4	44.4		
**Rational use of antibiotics**																								
Inadequate	23	34.8	6	9.2	**13.976 (0.001)**	**5.86 (2.16–15.89)**	16	50	2	5	**24.891 (<0.001)**	**26.18 (5.19–131.99)**	3	18.8	0	0	**4.285 (0.038)**	----	1	12.5	0	0	2.015 (0.365)	-----
Appropriate	34	51.5	52	80			11	34.4	36	90			13	81.3	21	100			7	87.5	8	88.9		
NV	9	13.6	7	10.8			5	15.6	2	5			0	0	0	0			0	0	1	11.1		

NV = not valuable. Source: database. Elaborated by the authors.

**Table 4 antibiotics-11-01829-t004:** Comparison of antibiotic consumption before and after training in empirical and targeted treatment of 257 patients of the internal medicine ward and intensive care unit of HEEE.

	Empirical Antibiotic Treatment			Targeted Antibiotic Treatment		
	Before	After	Student’s T (*p*)	Before	After	Student’s T (*p*)
**Bacteremia**						
Mean (SD)	2.07 (1.16)	1.81 (1.18)	0.207	2.19 (1.50)	1.53 (1.37)	0.01
**Pneumonia**						
Mean (SD)	1.87 (1.09)	1.05 (1.13)	0.003	2.15 (0.98)	2.05 (1.18)	0.618
**Urinary Infections**						
Mean (SD)	1.06 (0.99)	1.28 (1.00)	0.507	1.18 (1.27)	1.00 (1.14)	0.641
**Skin and Soft Tissue Infection**						
Mean (SD)	1.75 (0.88)	1.77 (0.66)	0.942	0.25 (0.70)	1.22 (1.20)	0.064

Source: database. Elaborated by the authors.

## Data Availability

The datasets generated and/or analyzed during the current study are not publicly available due to participant privacy but are available from the corresponding author upon reasonable request.

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
