# Peer review of "E-Learning versus Face-to-Face Methodology for Learning Antimicrobial Resistance and Prescription Practice in a Tertiary Hospital of a Middle-Income Country"

_antibiotics, 2022, doi:10.3390/antibiotics11121829_

Round 1

Reviewer 1 Report

This is an interventional study to improve knowledge on AMR using two different approaches - the  traditional teaching and the E-learning. The study is well designed and results well presented. The manuscript would benefit from further editing based on the comments below

  1. Abstract : 91 participants were physicians? In the text there are not only physicians but medical students as well 2. Introduction: Similar to the above. Please refere to existing data on medical students training on AMR ( Reference   Efthymiou P, Gkentzi D, Dimitriou G. Knowledge, Attitudes and Perceptions of Medical Students on Antimicrobial Stewardship. Antibiotics (Basel). 2020 Nov 17;9(11):821. doi: 10.3390/antibiotics9110821. PMID: 33213047; PMCID: PMC7698472.) 3. Methods: -  How was the randomisation occurred ?                     -  Were different disciplines equally represented? Would be good to show the number from each participating group ( ie...physicians;ICU, students etc)                     -  Tools used: Were the questionnaires the same for all participating groups                     -  Was there a pilot phase of the study? 4. Results  - Would be advisable to show the results by discipline ie in differents groups ( students, physicians, icu...) 5. Discussion: - As above comment- please comment on different disciplines                       - Expand on limitations: country specific, cultural differences, small number of participants, during COVID-19                       - It is also important to know that any intervention should be assessed for sustainability and long term outcomes                       - It would be advisable to authors to avoid any firm conclusions about the validity of such interventions and use words such as may, might  etc

Author Response

Dear reviewer 1,

Thank you very much for such valuable contributions for our work.

We answered all the comments and made the required changes on the manuscript.

Rev: This is an interventional study to improve knowledge on AMR using two different approaches - the traditional teaching and the E-learning. The study is well designed, and results well presented. The manuscript would benefit from further editing based on the comments below

RESPONSE TO THE REVIEWER 1

  1. Abstract :

Rev: 91 participants were physicians? In the text there are not only physicians but medical students as well.

Authors´ answer: The 91 participants are physicians with national medical licensing (all medical graduates): 29 specialists, 21 residents (general practitioner physician under training in hospital) and 41 postgraduates (Lato sensu) training for specialties. Of the 91, 49 belong to Internal Medicine and 42 to intensive care.

In Ecuador, there is no specialization in infectious diseases, who assume this role are internists. During the Covid 19 pandemic, the country manifested great deficiencies in health care, in hospital infrastructure, equipment and of course medical personnel. In Ecuador there is no professional qualification exam on antibiotic prescription and continuous training on this subject for practicing professionals is non-existent

  1. Introduction:

Rev: Similar to the above. Please refere to existing data on medical students training on AMR ( Reference   Efthymiou P, Gkentzi D, Dimitriou G. Knowledge, Attitudes and Perceptions of Medical Students on Antimicrobial Stewardship. Antibiotics (Basel). 2020 Nov 17;9(11):821. doi: 10.3390/antibiotics9110821. PMID: 33213047; PMCID: PMC7698472.) 

Authors´ answer: In Ecuador, doctors after finishing their undergraduate studies do a year of rural medicine, after which they receive the endorsement (medical license) to act as general practitioners. Resident physicians are general practitioners working in hospitals, postgraduate physicians are general practitioners who are specializing.

  1. Methods:

3.1. Rev: How was the randomization occurred?  

Authors´ answer:  Randomization was performed in Excel using random numbers, the list of all enrolled participants from internal medicine and intensive care was used. In this way, the members of both services were equally represented in each group.                 

3.2.  Rev: Were different disciplines equally represented? Would be good to show the number from each participating group ( ie...physicians; ICU, students etc)  

Authors´ answer: When performing the randomization in group 1 there were: 15 medical specialists, 12 residents and 20 postgraduates. In group 2 there were 14 specialists, 9 residents and 21 postgraduates. Regarding the units to which the participants belonged, 49 were internal medicine and 42 intensive care.                

3.3. Rev:  Tools used: Were the questionnaires the same for all participating groups        

 Authors´ answer: Yes, the questionnaires were the same for all participating groups.           

 3.4. Rev:  Was there a pilot phase of the study? 

Authors´ answer: A pilot test was carried out on 30 subjects, a Cronbach's alpha of 0.79 was obtained. We included this information in the manuscript. Thanks for reminding us about it.

  1. Results  -

  • Would be advisable to show the results by discipline ie in differents groups ( students, physicians, icu...)

Authors´ answer: Thanks for this suggestion. Actually, we considered to do this, but the sample is small (91 participants), and when divided by position  and service, the data became less representative. In addition, we needed to demonstrate the results in a homogeneous group, since in the future training would be given to all hospital staff, as residents, postgraduates and specialists evaluate and prescribe antimicrobials.

  1. Discussion:

5.1. Rev:- As above comment- please comment on different disciplines                      

Authors´ answer: See comment above

5.2. Rev:- Expand on limitations: country specific, cultural differences, small number of participants, during COVID-19 

Authors´ answer: Thanks for the suggestion to expand on limitations. This is very important and we added that to the main manuscript.

5.3. Rev: It is also important to know that any intervention should be assessed for sustainability and long term outcomes        

Authors´ answer: I agree with your reflection, however, all the resident and postgraduate doctors who participated in our study rotated their service every 3 months, so the long-term evaluation was not carried out. Sustainability is another important issue and this program is now incorporated in the Hospital continual professional development program.

5.4.  Rev: It would be advisable to authors to avoid any firm conclusions about the validity of such interventions and use words such as may, might  etc.

Authors´ answer: We appreciate and will apply your recommendation.

Reviewer 2 Report

Understanding to what extent e-learning could supplement or replace face-to-face (F2F) learning in specific settings is highly relevant to ensure continuous education is provided in the most efficient and effective way. This is a small study in a single hospital, whereby a subgroup of physicians was allocated to two groups. The first group started with a F2F training on AMR, followed by an elearning module on antimicrobial prescription; the second group started with elearning on AMR, followed by a F2F training on antimicrobial prescription. The authors describe this as a cross-over design, but this is not correct; a cross-over implies the same interventions are provided, please correct. It can not be excluded that the sequence is relevant for the measurement of outcomes.

A major shortcoming is that this small study is not a proper non-inferiority study. The sample size calculation provided relates to a precision estimate only, and is likely to be well underpowered to assess non-inferiority. As such, not observing a difference in short-term self-reported KAP outcomes cannot be taken as evidence for non-inferiority; please avoid any mentions of non-inferiority or equivalence, and modify this to stating you did not observe a statistically significant difference. Another weakness is the limitation to short term outcomes only, as acknowledged by the authors, but why was no later follow up included?

The selection of patients for the evaluation of prescription practices is not specified, making it difficult to assess representativeness, comparability, and potential selection bias. How were they selected, over how much time, any sample size considerations? Also, did they provide informed consent?

Several of the before-after prescription results reflected in table 2 are somewhat surprising, and make it hard to understand the medical context. Eg that 92.4% of bacteraemia patients before the intervention, and 86.2% afterwards were considered to have an indication not to treat? What was the clinical outcome of untreated bacteriaemia? Or that following the intervention, correct coverage among pneumonia patients decreased? How do the authors reflect on worse prescription practices after the training?

Knowledge and attitudes on AMR seemed to be very high before the intervention already, suggesting less need for (continuous) training, please reflect/discuss on priorities. Also, table 1 shows that improvement of KAP scores was either not observed (5/12) or marginal (+1; 7/12), please modify conclusions. AMR training might not be a similar priority as training and support for prescription, different from the current formulation in the discussion.

Minor shortcomings include the need for more details on the type of profession involved, eg microbiology, paediatrics, pharmacy, etc. Also, please provide more details on how pre and post KAP were measured, as well as if/how a questionnaire was piloted. Were attitudes and practices observed, or only self-reported? Some more attention to detail is warranted in general, eg use of ‘,’ vs ‘.’ with numbers, and consistent truncating of number of decimals to a meaningful number. Further English editing would be useful too.

Author Response

Dear reviewer 2,

We would like to thank you very much for such valuable contributions for our work.

We answered all the comments and made the required changes on the manuscript.

Understanding to what extent e-learning could supplement or replace face-to-face (F2F) learning in specific settings is highly relevant to ensure continuous education is provided in the most efficient and effective way. This is a small study in a single hospital, whereby a subgroup of physicians was allocated to two groups. The first group started with a F2F training on AMR, followed by an elearning module on antimicrobial prescription; the second group started with elearning on AMR, followed by a F2F training on antimicrobial prescription.

  1. Rev: The authors describe this as a cross-over design, but this is not correct; a cross-over implies the same interventions are provided, please correct. It can not be excluded that the sequence is relevant for the measurement of outcomes.

Authors´ answer: Thanks for calling our attention to this. We understand what you mean and we already corrected that in the methods section and in the abstract.

  1. Rev: A major shortcoming is that this small study is not a proper non-inferiority study. The sample size calculation provided relates to a precision estimate only, and is likely to be well underpowered to assess non-inferiority. As such, not observing a difference in short-term self-reported KAP outcomes cannot be taken as evidence for non-inferiority; please avoid any mentions of non-inferiority or equivalence, and modify this to stating you did not observe a statistically significant difference.

Authors´ answer: We agree with this and already made changes in the manuscript.

  1. Rev: Another weakness is the limitation to short term outcomes only, as acknowledged by the authors, but why was no later follow up included?

Authors´ answer: This is a phD project in partnership between Universidad Central de Ecuador and University of São Paulo-Brazil and there is a timeline for the program, so we did not have enough time to conclude this long-term follow-up.

  1. Rev: The selection of patients for the evaluation of prescription practices is not specified, making it difficult to assess representativeness, comparability, and potential selection bias. How were they selected, over how much time, any sample size considerations? Also, did they provide informed consent?

Authors´ answer: To evaluate the practice of prescription, ALL cases of infection (that came from all doctors) whose microbiological reports were positive and therefore had reports of susceptibility were evaluated. This evaluation was performed for a full month before and after the intervention. Based on the positive microbiological reports, we proceeded to search the digital clinical history for empirical antibiotic treatment (before the report) and targeted treatment (after the microbiological report), as well as other relevant information for the evaluation of the infection. The institutional endorsement of the hospital was obtained for the review of medical records. Patient data was always kept anonymous.

  1. Rev: Several of the before-after prescription results reflected in table 2 are somewhat surprising and make it hard to understand the medical context. Eg that 92.4% of bacteraemia patients before the intervention, and 86.2% afterwards were considered to have an indication not to treat?

What was the clinical outcome of untreated bacteriaemia? Or that following the intervention, correct coverage among pneumonia patients decreased? How do the authors reflect on worse prescription practices after the training?

Authors´ answer: It is important to briefly describe the context in which the research is carried out: Ecuador is a country in which there is no training for infectious diseases doctors or microbiologists, the few specialists in these areas have specialized outside the country. The country has incipient legislation on the subject of ADR and does not have practical prescription guides, hospitals that rarely have a Stewardship program running.

In Ecuador there is also no professional qualification exam that includes the prescription of antimicrobials to start the professional practice. In this context, an intervention could generate profound changes.

Regarding Table 2, we deeply appreciate reading it as it allows us to realize that it requires more explanation in order not to fall into errors. Table 2 compares the empirical treatment before and after the intervention. In the particular case of bacteraemia, 5 of 66 patients did not receive empirical treatment, compared to 9 of 65 patients after the intervention; that is, these patients had the indication not to give empiric treatment. (7.6% before versus 13.8% after). They represent cases that were interpreted as contamination of the blood sample.

  1. Rev: Knowledge and attitudes on AMR seemed to be very high before the intervention already, suggesting less need for (continuous) training, please reflect/discuss on priorities. Also, table 1 shows that improvement of KAP scores was either not observed (5/12) or marginal (+1; 7/12), please modify conclusions. AMR training might not be a similar priority as training and support for prescription, different from the current formulation in the discussion.

Authors´ answer: We agree with your point, the self-assessed knowledge, attitudes and practices impress with high results, which is why the evaluation of the prescription practice observed in this study is so valuable and shows deficiencies, deficiencies that must be intervened with training. We also believe that the greatest depth of training should be given in the prescription of antimicrobials. Knowledge of the AMR is a prerequisite for the correct choice of the antimicrobial to be used in a given case, therefore, in the training of all personnel, greater emphasis will be given to the module of good prescribing practices.

  1. Rev: Minor shortcomings include the need for more details on the type of profession involved, eg microbiology, paediatrics, pharmacy, etc. Also, please provide more details on how pre and post KAP were measured, as well as if/how a questionnaire was piloted. Were attitudes and practices observed, or only self-reported? Some more attention to detail is warranted in general, eg use of ‘,’ vs ‘.’ with numbers, and consistent truncating of number of decimals to a meaningful number. Further English editing would be useful too.

Authors´ answer: All the participants are medical specialists, residents and postgraduates in internal medicine and intensive care services. We included this information in the manuscript.

The questionnaire was piloted and evaluated by 7 experts in the area. We added this to the text.

The same questionnaire was used to assess all participants before and after the intervention. In the first phase of the study, reported knowledge, atitudes, and practices were assessed through a self-administered questionnaire. In the second part of the study, an evaluation of the observed practice was carried out, evaluating all the clinical cases attended by the personnel of the two participating services.

Round 2

Reviewer 2 Report

Thank you for modifications and clarifications. 

Please check carefully for persisting suggestions of non-inferiority/equivalence in an underpowered study (eg lines 36, lines 321). 

Also, please re-check Table 1: e.g. a p-value of 1 for the knowledge difference after training between F2F/Elearning?

Table 2: change 'nope' to 'no' (but more English corrections indicated throughout)

Author Response

Dear reviewer 2,

We would like to thank you very much for such valuable contributions for our work.

We answered all the comments and made the required changes on the manuscript.

Rev1: Please check carefully for persisting suggestions of non-inferiority/equivalence in an underpowered study (eg lines 36, lines 321). 

Authors´ answer: We agree with this and already made changes in the manuscript.

Rev 2: Also, please re-check Table 1: e.g. a p-value of 1 for the knowledge difference after training between F2F/Elearning?

Authors´ answer: The value reported by SPSS is 1.000, therefore it was corrected in table 2.

Rev 3: Table 2: change 'nope' to 'no' (but more English corrections indicated throughout)

Authors´ answer: We corrected what was indicated and a language correction was made by an English speaker.